# Understanding Generalization in Recurrent Neural Networks

**Zhuozhuo Tu, Fengxiang He, Dacheng Tao**
UBTECH Sydney AI Centre, School of Computer Science, Faculty of Engineering
The University of Sydney
Darlington, NSW 2008, Australia
`zhtu3055@uni.sydney.edu.au`, `{fengxiang.he,dacheng.tao}@sydney.edu.au`

## Abstract

In this work, we develop the theory for analyzing the generalization performance of recurrent neural networks. We first present a new generalization bound for recurrent neural networks based on matrix 1-norm and Fisher-Rao norm. The definition of Fisher-Rao norm relies on a structural lemma about the gradient of RNNs. This new generalization bound assumes that the covariance matrix of the input data is positive definite, which might limit its use in practice. To address this issue, we propose to add random noise to the input data and prove a generalization bound for training with random noise, which is an extension of the former one. Compared with existing results, our generalization bounds have no explicit dependency on the size of networks. We also discover that Fisher-Rao norm for RNNs can be interpreted as a measure of gradient, and incorporating this gradient measure not only can tighten the bound, but allows us to build a relationship between generalization and trainability. Based on the bound, we theoretically analyze the effect of covariance of features on generalization of RNNs and discuss how weight decay and gradient clipping in the training can help improve generalization.

## 1 Introduction

The Recurrent Neural network (RNN) is a neural sequence model that has achieved the state-of-the-art performance on numerous tasks, including natural language processing (Yang et al., 2018; Mikolov & Zweig, 2012), speech recognition (Chiu et al., 2018; Graves, 2013) and machine translation (Wu et al., 2016; Kalchbrenner & Blunsom, 2013). Unlike feedforward neural networks, RNNs allow connections among hidden units associated with a time delay. Through these connections, RNNs can maintain a "memory" that summarizes the past sequence of inputs, enabling it to capture correlations between temporally distant events in the data.

RNNs are very powerful, and empirical studies have shown that they have a very good generalization property. For example, Graves (2013) showed that deep LSTM RNNs achieved a test error of 17.7% on TIMT phoneme recognition benchmark after training with only 462 speech samples. Despite of the popularity of RNNs in practice, their theory is still not well understood. A number of recent works have sought to shed light on the effective representational properties of recurrent networks trained in practice. For example, Oymak (2018) studied the state equation of recurrent neural networks and showed that SGD can efficiently learn the unknown dynamics from few observations under proper assumptions. Miller & Hardt (2019) tried to explain why feed-forward neural networks are competitive with recurrent networks in practice. They identified stability as a necessary condition and proved that stable recurrent neural networks are well approximated by feed-forward networks for the purpose of both inference and training by gradient descent. Despite of the impressive progress in understanding the training behavior of RNNs, there is no generalization guarantee in these works.

Understanding the generalization performance in machine learning has been a central problem for many years and revived in recent years with the advent of deep learning. One classical approach to proving generalization bound is via notions of complexity. For deep neural networks, numerous complexity measures have been proposed to capture the generalization behavior such as VC

dimension (Harvey et al., 2017) and norm-based capacity including spectral norm (Bartlett et al., 2017; Neyshabur et al., 2019), Frobenius norm (Neyshabur et al., 2015b;a; 2018) and $l_p$-path norm (Neyshabur et al., 2015b; Bartlett & Mendelson, 2002; Golowich et al., 2018). These existing norm-based complexity measures depend on the number of hidden units of the network explicitly and thus can not explain why neural networks generalize so well in practice, despite that they operate in an overparametrized setting (Zhang et al., 2017). Neyshabur et al. (2019) proved generalization bounds for two layer ReLU feedforward networks, which decreased with the increasing number of hidden unit in the network. However their results only applied to two layer ReLU networks and some specific experiments. More recently, a new generalization bound based on Fisher-Rao norm was proposed (Liang et al., 2017). This notion of Fisher-Rao norm is motivated by information geometry and has good invariance properties. But they proved the bound only for linear deep neural networks. There are also some works about the generalization of recurrent neural networks (Zhang et al., 2018; Chen et al., 2019; Allen-Zhu & Li, 2019). However these bounds also depend on the size of networks, which makes them vacuous for very large neural networks.

Our main contributions are summarized as follows.

- We define the Fisher-Rao norm for RNNs based on its gradient structure and derive new Rademacher complexity bound and generalization bound for recurrent neural networks based on Fisher-Rao norm and matrix 1-norm. In contrast to existing results such as spectral norm-based bounds, our bound has no explicit dependence on the size of networks.

- We prove a generalization bound for RNNs when training with random noises. Our bound applies to general types of noises and can potentially explain the effect of noise training on generalization of recurrent neural networks as demonstrated by our empirical results.

- We propose a new technique to decompose RNNs with ReLU activation into a sum of linear network and difference terms. As a result, each term in the decomposition can be treated independently and easily when estimating the Rademacher complexity. This decomposition technique can potentially be applied to other neural networks architectures such as convolutional neural networks, which might be of independent interest.

The remainder of this paper is structured as follows. We define the problem and notations in Section 2. The notion of Fisher-Rao norm for RNNs is introduced in Section 3.1. We prove the generalization bound for RNNs in Section 3.2, and the generalization bound for training with random noise is derived in Section 3.3. Section 3.4 gives a detailed analysis of our generalization bound. Finally we conclude and discuss future directions.

## 2 PRELIMINARIES

We focus on the vanilla RNNs with ReLU activation. Let $U \in R^{m \times d}$, $V \in R^{k \times m}$ and $W \in R^{m \times m}$ be the weight matrices. Given the input sequence $x = (x_1, x_2, \cdots, x_L) \in R^{Ld}$ where each $x_i \in R^d$ and $L$ is the input sequence length, the vanilla RNNs can be described as follows.

$$
\begin{aligned}
g_t &= U x_t + W h_{t-1}, \\
h_t &= \rho(g_t), \\
y_t &= V h_t,
\end{aligned}
\tag{1}
$$

where $g_t$ and $h_t \in R^m$ represents the input and output of hidden layer at step $t$, $\rho(\cdot)$ is the ReLU function and $y_t \in R^k$ denotes the output value at step $t$.

For simplicity, in this paper, we only consider the final output $y_L$. We assume that data $(x, y)$ is drawn i.i.d. from some unknown distribution $\mathcal{D}$ over $R^{Ld} \times \mathcal{Y}$ where $\mathcal{Y}$ represents the label space $\{1, 2, \cdots, k\}$. The RNNs above define a mapping $y_L(x)$ from $R^{Ld} \to R^k$, where $k$ is the number of classes. We convert $y_L(x)$ to a classifier by selecting the output coordinate with the largest magnitude, meaning

$$x \to \operatorname{argmax}_i [y_L(x)]_i,$$

where $[\cdot]_i$ represents the $i$-th element of a vector. This naturally leads to the definition of margin $\mathcal{M}_{y_L}(x, y)$ of the output $y_L$ at a labeled example $(x, y)$:

$$\mathcal{M}_{y_L}(x, y) = [y_L(x)]_y - \max_{y' \neq y} [y_L(x)]_{y'}.$$

Thus, $y_L$ misclassifies $(x, y)$ if and only if $\mathcal{M}_{y_L}(x, y) \leq 0$. The quality of the prediction made by $y_L$ is measured by the expected risk defined as

$$\mathbb{E}_{(x,y)\sim\mathcal{D}}[\mathbb{1}_{\mathcal{M}_{y_L}(x,y)\leq 0}].$$

Since the underlying distribution $\mathcal{D}$ is unknown to us, we instead consider the empirical error on sample data given by

$$\frac{1}{n}\sum_{i=1}^{n}(\mathbb{1}_{\mathcal{M}_{y_L}(x_i,y_i)\leq\alpha}).$$

The generalization error is then the difference between expected risk and empirical risk, defined as

$$\mathbb{E}_{(x,y)\sim\mathcal{D}}[\mathbb{1}_{\mathcal{M}_{y_L}(x,y)\leq 0}] - \frac{1}{n}\sum_{i=1}^{n}(\mathbb{1}_{\mathcal{M}_{y_L}(x_i,y_i)\leq\alpha}).$$

Our goal in this paper is to study the generalization error for RNNs theoretically.

To establish the generalization bound, a little bit of notations are necessary. For a vector, we denote the $l_p$ norm by $||v||_p = (\sum|v_i|^p)^{1/p}$ and the $l_\infty$ norm by $||v||_\infty = \max|v_i|$. For a matrix, we denote the matrix $p$-norm as $||A||_p = \max_{||x||_p=1}||Ax||_p$, the matrix 1-norm by $||A||_1 = \max_j\{\sum_i|a_{ij}|\}$ and the Frobenius norm by $||A||_F^2 = trace(AA^T)$. The smallest eigenvalue of a matrix A is given by $\lambda_{min}(A)$. The activation function $\rho$ and its derivative $\rho'$ are entrywise, i.e., $\rho(A) = (\rho(a_{ij}))_{ij}$ and $\rho'(v) = (\rho'(v_i))_i$. We denote $c = (L+1, L, \cdots, 2)^T$, $\eta(\theta) = [V diag(\rho'(g_L))...W diag(\rho'(g_1))Ux_1, V diag(\rho'(g_L))...W diag(\rho'(g_2))Ux_2, \cdots, V diag(\rho'(g_L)) Ux_L] \in R^{k\times L}$ and $\tau(\theta) = (VW^{L-1}Ux_1, VW^{L-2}Ux_2, \cdots, VUx_L)$ where $\theta = (U, W, V)$ and $diag$ converts a vector into a diagonal matrix.

## 3 MAIN RESULT

In this section, we prove a generalization bound for RNNs with ReLU activation. Our new bound is based on Fisher-Rao norm and matrix 1-norm. We first define the Fisher-Rao norm for RNNs.

### 3.1 FISHER-RAO NORM FOR RNNS

We adapt the notion of Fisher Rao norm to recurrent neural networks. To begin with, we establish the following structural result for RNNs.

**Lemma 1.** *Given an input $x = (x_1, x_2, \cdots, x_L)$, consider the recurrent neural network in (1), we have the identity*

$$\sum_{a,b}\frac{\partial y_L}{\partial v_{ab}}v_{ab} + \sum_{i,j}\frac{\partial y_L}{\partial w_{ij}}w_{ij} + \sum_{p,q}\frac{\partial y_L}{\partial u_{pq}}u_{pq} = \eta(\theta)c.$$

The notion of Fisher-Rao norm is motivated by Fisher-Rao metric of information geometry and is defined as follows.

**Definition 1** ((Liang et al., 2017), Definition 2). *The Fisher-Rao norm for a parameter $\theta$ is defined as*

$$||\theta||_{fr}^2 := < \theta, I(\theta)\theta >,$$

*where $I(\theta) = \mathbb{E}(\nabla l(y_{L\theta}(x), y) \otimes \nabla l(y_{L\theta}(x), y))$ and $l(.,.)$ is the loss function.*

The following lemma gives the explicit formula of Fisher-Rao norm in RNNs. We can see that the notion of Fisher-Rao norm relies mainly on the gradient structure of RNNs.

**Lemma 2.** *Assume that the loss function $l(.,.)$ is smooth in the first argument. Then the following identity holds for the RNN in (1),*

$$||\theta||_{fr}^2 = \mathbb{E}\big(\langle\eta(\theta)c, \frac{\partial l(y_{L\theta}(x), y)}{\partial y_{L\theta}}\rangle^2\big).$$

**Remark 1.** We observe that each term $V diag(\rho'(g_L))...W diag(\rho'(g_i))U x_i$ in $\eta(\theta)$ is actually the gradient component in Backpropagation through time (BPTT). Therefore, the Fisher-Rao norm can be regarded as a measure of gradient. As will be shown later, we can build a relationship between generalization and trainability in RNNs via Fisher-Rao norm.

For the linear activation function and margin loss $l(y_{L\theta}(x), y) = \Phi_\alpha(\mathcal{M}_{y_L}(x, y))$ where $\alpha > 0$ is the margin parameter, one might upper bound the Fisher-Rao norm in Lemma 2 by

$$||\theta||_{fr}^2 \leq \frac{4}{\alpha^2} \mathbb{E}\big( \max_i [(\tau(\theta)c)_i]^2 \big),$$

since $\big\langle \tau(\theta)c, \frac{\partial l(y_{L\theta}(x), y)}{\partial y_{L\theta}} \big\rangle^2 \leq \frac{4}{\alpha^2} \max_i [(\tau(\theta)c)_i]^2$ by definition of $\mathcal{M}_{y_L}(x, y)$ and lipschitz property of $\Phi_\alpha(\cdot)$. We define this upper bound as

$$||\theta||_{fs}^2 := \mathbb{E}\big( \max_i [(\tau(\theta)c)_i]^2 \big), \tag{2}$$

and still call it "Fisher-Rao norm" in the paper by slightly abusing the terminology as they are equivalent for $k = 1$. In the rest of the paper, we will use this Fisher-Rao norm $|| \cdot ||_{fs}$ to derive generalization bound for RNNs.

### 3.2 GENERALIZATION BOUND FOR RNNS

We use matrix 1-norm and Fish-Rao norm together to derive a generalization bound for RNNs. Since it is very challenging to bound the Radermacher complexity of ReLU networks directly in terms of the Fisher-Rao norm, we consider decomposing the ReLU network into the sum of a linear network and a difference term, i.e., $y_L = \psi(\theta)x + (y_L - \psi(\theta)x)$. For the linear network part $\psi(\theta)x$, the Rademacher complexity can be bounded directly by Fisher-Rao norm. For the difference term $(y_L - \psi(\theta)x)$, we further decompose it into a sum of simpler terms and then upper bound the Rademacher complexity of these simpler terms by matrix 1-norm. We first give the results for the linear network part.

**Lemma 3.** *Define* $\mathcal{F}_r := \{x \rightarrow [\psi(\theta)x]_y : ||\theta||_{fs} \leq r, y \in \mathcal{Y}\}$ *where* $x \in R^{Ld}$ *and* $\psi(\theta) := (VW^{L-1}U, VW^{L-2}U, \cdots, VU)$. *For any data* $x_1, x_2, \cdots, x_n$ *drawn i.i.d from the distribution* $\mathcal{D}$, *collect them as columns of a matrix* $X \in R^{Ld \times n}$. *Then we have*

$$\hat{\mathfrak{R}}_n(\mathcal{F}_r) \leq \frac{r||X||_F}{2n} \sqrt{\frac{1}{\lambda_{min}(\mathbb{E}(xx^T))}},$$

*assuming that* $\mathbb{E}(xx^T)$ *is positive definite.*

**Remark 2.** If $\mathbb{E}(x) = 0$, $\mathbb{E}(xx^T)$ is the covariance matrix of random variable $x$.

**Remark 3.** We should mention that our assumption that $\mathbb{E}(xx^T)$ is positive definite is not so restrictive and usually holds in practice. For example, for the case that $x$ is continuous random variable, we can prove that $E(xx^T)$ is positive definite as follows. Suppose that $x$ is a continuous random variable in the $n$-dimensional subspace $X \subset R^n$. If there exists $u \in R^n$ such that $u^T E(xx^T)u = 0$, then for any $x \in X$ we have $u^T x = 0$, i.e., $u \perp X$. Since $X$ is $n$-dimensional, the only $u$ that satisfies is that $u = 0$. Therefore, by definition, $E(xx^T)$ is positive definite. As we will show in Section 3.3, this assumption can be removed, and a more general generalization bound will be presented.

Now we bound the Rademacher complexity of the difference term $y_L - \psi(\theta)x$. With a slight abuse of notations, given input data $x_1, x_2, \cdots, x_n \in R^{Ld}$, the corresponding $g_1, g_2, \cdots, g_n \in R^{Lm}$ and $h_1, h_2, \cdots, h_n \in R^{Lm}$ are calculated by (1). We collect all input data as a matrix denoted by $X$, all input data at time $t$ as a matrix denoted by $X_t$, all input of the hidden layer at time $t$ as a matrix denoted by $G_t$ and all output of hidden layer at time $t$ denoted by $H_t$, where $X \in R^{Ld \times n}$, $X_t \in R^{d \times n}, G_t \in R^{m \times n}$, $H_t \in R^{m \times n}$ and $t = 1, \cdots, L$. The difference term can be decomposed by the following lemma.

**Lemma 4.** *Define* $H_t'' := H_t - G_t$. *Then the following equality holds*

$$VH_L - \psi(\theta)X = \sum_{i=1}^{L} VW^{L-i}H_i''.$$

To bound the Rademacher complexity of each term in the above decomposition, we need a technical lemma given as follows.

**Lemma 5.** *For any $p \geq 1$, $||H_t''||_p \leq m^{\frac{1}{p}(1-\frac{1}{p})} n^{\frac{1}{p}(1-\frac{1}{p})} ||G_t||_p$.*

As we will see, the operator norm in Lemma 5 will be instantiated for the case of $p = 1$. The use of $||\cdot||_1$ helps avoid the appearance of the dimension $m$ when upper bounding the Rademacher complexity. Also it guarantees that Rademacher complexity has a convergence rate $\mathcal{O}(1/n)$. The upper bound for the Rademacher complexity of these individual term is given by the following lemma.

**Lemma 6.** *Let $\Omega := \{\theta = (U, W, V) : ||V^T||_1 \leq \beta_V, ||W^T||_1 \leq \beta_W, ||U^T||_1 \leq \beta_U\}$. Then for any $i = 1, \cdots, L$, we have*

$$\mathbb{E}_\sigma \Big( \sup_{\theta \in \Omega, y \in \mathcal{Y}} \frac{1}{n} [VW^{L-i}H_i'']_y, \sigma \Big) \leq \frac{1}{n} \beta_V \beta_U \sum_{j=1}^i \beta_W^{L-j} ||X_j^T||_1,$$

*where $\sigma = (\sigma_1, \sigma_2, \cdots, \sigma_n)^T$ is Rademacher random variable and $[\cdot]_y$, represents the y-th row of the matrix.*

We are now ready to put the ingredients together to prove our first theorem.

**Theorem 1** (Rademacher complexity of RNNs). *Let $\overline{\Omega} := \{\theta = (U, W, V) : ||V^T||_1 \leq \beta_V, ||W^T||_1 \leq \beta_W, ||U^T||_1 \leq \beta_U, ||\theta||_{fs} \leq r\}$. Then, the empirical Rademacher complexity of RNNs with ReLU can be bounded as follows*

$$\mathbb{E}_\sigma \Big( \sup_{\theta \in \overline{\Omega}, y \in \mathcal{Y}} \frac{1}{n} \sum_{i=1}^n [y_{L\theta}(x_i)]_y \sigma_i \Big) \leq \frac{r||X||_F}{2n} \sqrt{\frac{1}{\lambda_{min}(\mathbb{E}(xx^T))}} + \frac{1}{n} \beta_V \beta_U ||X^T||_1 \Lambda ,$$

*where $\Lambda := \frac{1}{1-\beta_W} \Big( \frac{1-\beta_W^L}{1-\beta_W} - L\beta_W^L \Big)$ if $\beta_W \neq 1$ and $\Lambda := \frac{L+L^2}{2}$ for $\beta_W = 1$.*

To establish the generalization bound for RNNs, we need the following classical results for multi-class margin bounds.

**Lemma 7** ((Kuznetsov et al., 2015), Theorem 2). *Let $H \subseteq \mathbb{R}^{\mathcal{X} \times \mathcal{Y}}$ be a hypothesis set with $\mathcal{Y} = \{1, 2, \cdots, k\}$. Fix $\alpha > 0$. Then, for any $\delta > 0$, with probability at least $1 - \delta$, the following multi-class classification generalization bound holds for all $h \in H$:*

$$R(h) \leq \frac{1}{n} \sum_{i=1}^n \Phi_\alpha(\mathcal{M}_h(x_i, y_i)) + \frac{4k}{\alpha} \hat{\mathfrak{R}}_n(\Pi_1(H)) + 3\sqrt{\frac{\log \frac{2}{\delta}}{2n}},$$

*where $\Pi_1(H) = \{x \to h(x, y) : y \in \mathcal{Y}, h \in H\}$.*

The generalization bound for RNNs follows from combining Theorem 1 and Lemma 7.

**Theorem 2.** *Fix margin parameter $\alpha$, then for any $\delta > 0$, with probability at least $1 - \delta$, the following holds for every RNN whose weight matrices $\theta = (U, W, V)$ satisfy $||V^T||_1 \leq \beta_V, ||W^T||_1 \leq \beta_W, ||U^T||_1 \leq \beta_U$ and $||\theta||_{fs} \leq r$:*

$$\mathbb{E}[\mathbb{1}_{\mathcal{M}_{y_L}(x,y) \leq 0}] \leq \frac{1}{n} \sum \mathbb{1}_{\mathcal{M}_{y_L}(x_i, y_i) \leq a} + \frac{4k}{\alpha} \Big( \frac{r||X||_F}{2n} \sqrt{\frac{1}{\lambda_{min}(\mathbb{E}(xx^T))}} + \frac{1}{n} \beta_V \beta_U ||X^T||_1 \Lambda \Big) +$$
$$3\sqrt{\frac{\log \frac{2}{\delta}}{2n}}$$

$$\text{(3)}$$

**Comparison with existing results.** We compare our result with the existing generalization bounds (Zhang et al., 2018; Chen et al., 2019). In comparison with the bound in Zhang et al. (2018), which is of the order $\tilde{\mathcal{O}}\Big( \frac{\max\{d, m, k\} L^2 ||U||_2 ||V||_2 \max\{1, ||W||_2^L\}}{\sqrt{n}\alpha} \Big)$, there is no explicit appearance of the network size parameters $d$ and $m$ in our bound. As we have mentioned before, the reason that we can avoid these dimensional factors is that we use matrix 1-norm instead of spectral norm to

upper bound the Rademacher complexity of the network. Moreover, there is always a $L^2$ factor in their bound, whereas the $L^2$ term only occurs in our bound when $||W^T||_1 = 1$. For the case that $||W^T|| > 1$, our bound has only a linear dependence on $L$, and for the case that $||W^T||_1 < 1$, by simple calculation, we can show that $\Lambda \leq \frac{1}{(1-\beta_W)^2}$ and the dependence on $L$ would vanish. Both of our bounds have an exponential term $||W||^L$, which would make the bounds become vacuous for $||W|| > 1$. It should also be pointed out that our bound scales linearly with the number of classes since we handle multiclass on each coordinate of a $k$-tuple of functions and pay a factor of $k$. Chen et al. (2019) also derived generalization bound for RNNs in terms of spectral norm and the total number of parameters of the network by using covering number analysis. Since their work assumed that the activation function in the hidden layers was bounded rather than the ReLU activation function considered in our paper, their bound is not directly comparable to ours, and we do not make a comparison here due to the page limit. We should emphasis that our proof technique is totally different from the PAC-Bayes approach (Zhang et al., 2018) and covering number analysis (Chen et al., 2019). In particular, we work on the Rademacher complexity of RNNs directly with no invocation of complicated tools such as covering number, which makes our analysis conceptually much simpler. There is also an additional bonus of our proof technique. In the next section, we will use this proof technique to derive a generalization bound for RNNs when training with random noise.

### 3.3 Generalization Bound for Training with Random Noise

The generalization bound in Theorem 2 requires the input covariance matrix $\mathbb{E}(xx^T)$ to be positive definite and would become very poor when the smallest eigenvalue is close to 0, which greatly limits the power of our bound. To address this issue, we consider adding random noise to the input data. We notice that after adding random noise with mean 0 and variance $\sigma_\epsilon^2$, the term $\mathbb{E}(xx^T)$ in the bound becomes $\mathbb{E}((x + \epsilon)(x + \epsilon)^T)$ and the smallest eigenvalue of $\mathbb{E}((x + \epsilon)(x + \epsilon)^T)$ is $(\lambda_{min}(\mathbb{E}(xx^T)) + \sigma_\epsilon^2)$, which is greater than $\sigma_\epsilon^2$. Therefore, our bound is still applicable even when the covariance matrix of original input data is rank-deficient. Involving noise variables has been widely used in recurrent neural networks as a regularization technique (Bayer et al., 2013; Zaremba et al., 2014; Dieng et al., 2018; Gal & Ghahramani, 2016). For example, Bayer et al. (2013) claimed that conventional dropout did not work well with RNNs because the recurrence amplified noise, which in turn hurt learning. To fix this problem, Zaremba et al. (2014) proposed to inject noise only to the input and output of RNNs. Although their method greatly reduced overfitting on a variety of tasks, the generalization guarantee was not provided. In this section, we present a generalization bound for RNNs with noise training. For simplicity, we assume that the noise is drawn $i.i.d.$ from a Gaussian distribution with zero mean and variance $\sigma_\epsilon^2$. Let $\epsilon_i$ denotes the $d$-dimensional gaussian noise generated at step $i$ and $\epsilon = (\epsilon_1, \epsilon_2, \cdots, \epsilon_L) \in R^{Ld}$. We collect all noise data as a matrix denoted by $X_\epsilon$. To prove the generalization bound, we need to use the Lipschitz property of RNNs given by the following lemma.

**Lemma 8.** *For every RNN in (1) with weight matrices $\theta = (U, W, V)$, $y_L$ is Lipschitz with respect to $|| \cdot ||_\infty$, i.e.,*

$$||y_L(x) - y_L(x')||_\infty \leq \sum_i ||V^T||_1 ||U^T||_1 ||W^T||_1^{L-i} ||x_i - x_i'||_\infty$$

*for any $x = (x_1, x_2, \cdots, x_L), x' = (x_1', x_2', \cdots, x_L') \in R^{Ld}$.*

The generalization bound for training with random noise is described as follows.

**Theorem 3.** *Fix margin parameter $\alpha$, then for any $\delta > 0$, with probability at least $1 - \delta$ over a sample $((x_1, \epsilon_1, y_1), (x_2, \epsilon_2, y_2), \cdots, (x_n, \epsilon_n, y_n))$, the following holds for every RNN whose weight matrices $\theta = (U, W, V)$ satisfy $||V^T||_1 \leq \beta_V, ||W^T||_1 \leq \beta_W, ||U^T||_1 \leq \beta_U$ and $||\theta||_{fs} \leq r$:*

$$\mathbb{E}[\mathbb{1}_{\mathcal{M}_{y_L}(x,y) \leq 0}] \leq \frac{1}{n} \sum \Phi_\alpha(\mathcal{M}_{y_L}(x_i + \epsilon_i, y_i)) + \frac{2}{\alpha} \sum_i \beta_V \beta_U \beta_W^{L-i} \sigma_\epsilon \sqrt{2\log(2d)} + 3\sqrt{\frac{\log\frac{2}{\delta}}{2n}} +$$
$$\frac{4k}{\alpha}\left(\frac{r||X + X_\epsilon||_F}{2n}\sqrt{\frac{1}{\lambda_{min}(\mathbb{E}(xx^T)) + \sigma_\epsilon^2}} + \frac{1}{n}\beta_V \beta_U ||X^T + X_\epsilon^T||_1 \Lambda\right)$$

.

**Remark 4.** The above bound can be easily extended to other kinds of noises by replacing $\sigma_\epsilon \sqrt{2\log(2d)}$ by $\mathbb{E}_\epsilon ||\epsilon_i||_\infty$.

**Remark 5.** The bound in Theorem 3 is an extension of that in Theorem 2 and can be applied even when the smallest eigenvalue of $\mathbb{E}(xx^T)$ is very close to 0. For example, when $\lambda_{min}(\mathbb{E}(xx^T)) = 1 \times 10^{-6}$, applying Theorem 2 directly might lead to a vacuous bound. But if using Theorem 3 by choosing a small noise with mean 0 and variance 0.01, we might obtain a better bound since the term $\sqrt{\frac{1}{\lambda_{min}(\mathbb{E}((xx^T)))+\sigma_\epsilon^2}} \leq 10$. Notice that adding noise can not always guarantee an improved generalization especially when $\lambda_{min}(\mathbb{E}(xx^T))$ is not so small as it incurs an additional linear term $\frac{2}{\alpha}\sum_i \beta_V \beta_U \beta_W^{L-i} \sigma_\epsilon \sqrt{2\log(2d)}$ and might also increase other parameters in the bound such as $||X + X_\epsilon||_F$. Therefore we suggest adding noise only when the smallest eigenvalue of $\mathbb{E}(xx^T)$ is very small. For this case, a small noise such as $\sigma_\epsilon = 0.1$ not only can greatly improve the term $\sqrt{\frac{1}{\lambda_{min}(\mathbb{E}(xx^T))}}$ but also ensure that the extra cost $\sigma_\epsilon \sqrt{2\log(2d)}$ and $||X + X_\epsilon||_F/n$ be small enough since $||X + X_\epsilon||_F/n \leq ||X||_F/n + ||X_\epsilon||_F/n$ and $||X_\epsilon||_F/n$ would be small when $n$ is large.

**Remark 6.** If we remove the constraint condition $||\theta||_{fs} \leq r$, which means that we do not have any knowledge about the gradients, the generalization bound in Theorem 2 and Theorem 3 still holds by substituting $r$ with $\beta_V \beta_U B(\frac{1}{(1-\beta_W)^2} + \frac{1}{1-\beta_W})$ for $\beta_W < 1$. But with this extra gradient measure, the bound can become much tighter, especially when $\lambda_{min}(\mathbb{E}(xx^T)$ is small. Please refer to the detailed analysis in the next section.

**Experiments.** We now study the effect of random noise on generalization of RNNs empirically. For simplicity, we consider the IMDB dataset, a collection of 50K movie reviews for binary sentiment classification. We use GloVe word embedding to map each word to a 50-dimensional vector. We train vanilla RNNs with ReLU activation function for sequence length $L = 100$. The corresponding smallest eigenvalue of $\mathbb{E}(xx^T)$ is approximated by using the total training data, which is $4 \times 10^{-4}$. We add Gaussian noise to the input data in the training process with $\sigma_\epsilon = 0.1, 0.2, 0.3$ and $0.4$. The generalization error which is the gap between test error without noise and training error with noise for $L = 100$ and different values of $\sigma_\epsilon$ is shown in Figure 1 (results for other values of $L$ in Appendix D). We observe that as we start injecting noise, the generalization error becomes better. But when the deviation

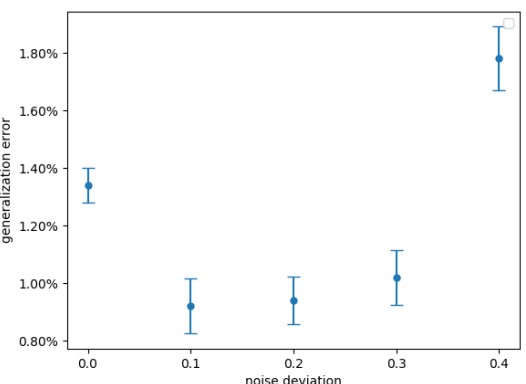

Figure 1: Generalization error for training with noise (mean $\pm$ standard error averaged on 5 runs).

of noise keeps growing, the generalization error shows an increasing tendency. This behavior is consistent with the prediction made by our bound.

## 3.4 ANALYSIS OF GENERALIZATION BOUND

Our theoretical results have a number of implications for the generalization performance in RNNs, and some of them have been observed in empirical studies. We summarize these implications as follows.

### 3.4.1 GENERALIZATION AND SMALLEST EIGENVALUE OF $\mathbb{E}(xx^T)$

According to our results, the generalization performance in RNNs is influenced by the smallest eigenvalue of $\mathbb{E}(xx^T)$. Since the smaller eigenvalues may contribute to high frequency components of the input signal, our bound suggests that high frequency information is potentially more difficult to generalize, which is consistent with intuition. There are many factors that impact on the smallest eigenvalue and therefore the generalization performance in RNNs. In particular, we study the effect of the correlation between features on the generalization in RNNs. The exact answer for this problem may be complicated. Here we only make an initial attempt and claim that weaker correlation would

help improve the generalization, and a non-rigorous proof is given as follows. Denote the covariance matrix $\mathbb{E}(xx^T)$ by $\Xi$ where each element $\xi_{ij}$ in $\Xi$ represents the covariance between feature $i$ and $j$. Suppose that $||\Xi - I||_1 \leq \zeta$ with $\zeta < 1$. By definition of $|| \cdot ||_1$ matrix norm, we immediately get $|\xi_{ii} - 1| + \sum_{j \neq i} |\xi_{ij}| \leq \zeta$ for any $i$. Then by simple derivation, we obtain $\xi_{ii} - \sum_{j \neq i} |\xi_{ij}| \geq 1 - \zeta$ for any $i$. Applying Gershgorin circle theorem, we have that the smallest eigenvalue must be greater or equal than $1 - \zeta$. Since the element $\xi_{ij}$ with $i \neq j$ represents the covariance between feature $i$ and $j$, a weaker correlation between feature $i$ and $j$ means a smaller value of $|\xi_{ij}|$ and we need a smaller $\xi$ to upper bound $||\Xi - I||_1$, which gives us a bigger lower bound on the smallest eigenvalue. Therefore the generalization bound becomes better.

### 3.4.2 GENERALIZATION AND TRAINABILITY

The generalization of RNNs also depends on parameters $\beta_U, \beta_V, \beta_W$ and $r$, where $\beta_U, \beta_V$ and $\beta_W$ control the weight matrices and $r$ represents the gradient measure. It has a natural relationship with the training process. The normal procedure in training RNNs is to use weight decay for regularization and gradient clipping to avoid the exploding gradients problems (Bengio et al., 1994; Pascanu et al., 2013). From the perspective of generalization, these strategies can decrease the value of these parameters $\beta_U, \beta_V, \beta_W$ and $r$ and thus improves the generalization. For example, if $\beta_W \leq 1$, we have $\Lambda \leq \frac{1}{(1 - \beta_W)^2}$, and the second term $\frac{1}{n} \beta_V \beta_U ||X^T||_1 \Lambda$ in the generalization bound would be small when $\beta_W$ is not so close to 1. Similarly, if $\lambda_{min}(\mathbb{E}(xx^T))$ is very small, by setting the gradient clipping value in the training procedure, we can achieve a smaller value of $r$ and thus good generalization. Therefore our bound partially explains why training RNNs in this way can achieve good performance in practice.

### 3.4.3 GENERALIZATION AND GRADIENT MEASURE

We are interested in how the gradient measure contributes to generalization. Suppose now that we only have the weights, i.e., the parameters $\beta_U, \beta_W$ and $\beta_V$ and the gradient measure parameterized by $r$ is unknown to us. To apply our bound, a natural idea is to infer the gradient measure parameter $r$ based on the known weight parameters. An upper bound for $r$ in terms of $\beta_U, \beta_W$ and $\beta_V$ is given as follows. Under the same conditions as Theorem 2, if we further assume that the data $x$ be given with $||x^T||_1 \leq B$, by the definition of $|| \cdot ||_{fs}$ in (2), for any $y \in \mathcal{Y}$, we have

$$
\begin{aligned}
\left((\tau(\theta)c)_y\right)^2 &= ((L+1)[V]_y, W^{L-1}Ux_1 + L[V]_y, W^{L-2}Ux_2 + \cdots + 2[V]_y, Ux_L)^2 \\
&\leq (|(L+1)[V]_y, W^{L-1}Ux_1| + |L[V]_y, W^{L-2}Ux_2| + \cdots + |2[V]_y, Ux_L|)^2 \\
&\leq ((L+1)\beta_V\beta_U B\beta_W^{L-1} + L\beta_V\beta_U B\beta_W^{L-2} + 2\beta_V\beta_U B)^2 \\
&= (\beta_V\beta_U B(\tfrac{\beta_W - \beta_W^L}{(1 - \beta_W)^2} + \tfrac{2 - (L+1)\beta_W^L}{1 - \beta_W}))^2 \leq (\beta_V\beta_U B(\tfrac{1}{(1 - \beta_W)^2} + \tfrac{1}{1 - \beta_W}))^2
\end{aligned},
$$

for $\beta_W < 1$, and $\left((\tau(\theta)c)_y\right)^2 \leq (\beta_V\beta_U B\frac{3L+L^2}{2})^2$ for $\beta_W = 1$. The above inequality holds for any $x$ and $y$. So we can get $||\theta||_{fs} = \mathbb{E}\left(\max_i[(\tau(\theta)c)_i]^2\right)^{1/2} \leq \beta_V\beta_U B(\frac{1}{(1 - \beta_W)^2} + \frac{1}{1 - \beta_W})$ for $\beta_W < 1$. By replacing $r$ with $\beta_V\beta_U B(\frac{1}{(1 - \beta_W)^2} + \frac{1}{1 - \beta_W})$, the generalization bound (3) also holds. But notice that this bound is obtained without any knowledge about the gradients. If we happen to know that the parameter $r$ is much smaller than $\beta_V\beta_U B(\frac{1}{(1 - \beta_W)^2} + \frac{1}{1 - \beta_W})$, for example, by setting gradient clipping value to be small in training process, this extra gradient measure can provide us with a better generalization bound, especially when the smallest eigenvalue of $\mathbb{E}(xx^T)$ is small. Therefore the introduction of Fisher-Rao norm can help eliminate the negative effect of $\lambda_{min}(\mathbb{E}(xx^T))$ and thus improve the generalization bound.

## 4 CONCLUSION

In this paper, we propose a new generalization bound for RNNs in terms of matrix 1-norm and Fisher-Rao norm, which has no explicit dependence on the size of networks. Based on the bound, we analyze the influence of covariance of features on generalization of RNNs and discuss how weight decay and gradient clipping in the training can help improve generalization. While our bound is useful for analyzing generalization performance of RNNs, it would become vacuous because of an exponential term when $||W^T||_1 > 1$. It is of interest to get a tighter bound which can avoid this

exponential dependence. Moreover, our bound only applies to vanilla RNNs with ReLU activation, and extending the results to tangent and sigmoid activation functions or other variants of RNNs like LSTM and MGU might be an interesting topic for future research.

ACKNOWLEDGMENTS

This work was supported by Australian Research Council Project FL-170100117. We would like to thank Gemeng Zhang from Tulane University for helpful discussions and all the reviewers for their constructive comments.

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

## A  PROOFS IN SECTION 3.1

### A.1  PROOF OF LEMMA 1

*Proof.* To begin with, by equation (1), we have $y_L = V h_L$. Then the derivative of $y_L$ with respect to $v_{ab}$ can be calculated as

$$\frac{\partial y_L}{\partial v_{ab}} = (0, 0, \cdots, [h_L]_b, \cdots, 0)^T,$$

i.e., a $k$-dimensional vector with $a$-th element $[h_L]_b$ and all other elements zero. Multiplying both sides by $v_{ab}$ and summing them up, we get

$$\sum_{a,b} \frac{\partial y_L}{\partial v_{ab}} v_{ab} = V h_L = y_L.$$

The derivative of $y_L$ with respect to $W$ and $U$ can be derived by using chain rule in the similar way as follows.

$$\begin{aligned}
\frac{\partial y_L}{\partial w_{ij}} &= V \frac{\partial h_L}{\partial w_{ij}} \\
\frac{\partial h_L}{\partial w_{ij}} &= diag(\rho'(g_L))(0, \cdots, [h_{L-1}]_j, \cdots, 0)^T + diag(\rho'(g_L)) W \frac{\partial h_{L-1}}{\partial w_{ij}}
\end{aligned}$$

and

$$\begin{aligned}
\frac{\partial y_L}{\partial u_{pq}} &= V \frac{\partial h_L}{\partial u_{pq}} \\
\frac{\partial h_L}{\partial u_{pq}} &= diag(\rho'(g_L))(0, \cdots, [x_L]_q, \cdots, 0)^T + diag(\rho'(g_L)) W \frac{\partial h_{L-1}}{\partial u_{pq}}
\end{aligned} \quad,$$

where we use the property of ReLU activation function that $\rho(z) = \rho'(z)z$. Summing up these terms immediately gives us the following equality.

$$\begin{aligned}
&\sum_{i,j} \frac{\partial h_L}{\partial w_{ij}} w_{ij} + \sum_{p,q} \frac{\partial h_L}{\partial u_{pq}} u_{pq} \\
&= diag(\rho'(g_L))(\sum_{i,j}(0, \cdots, [h_{L-1}]_j, \cdots, 0)^T w_{ij} + \sum_{p,q}(0, \cdots, [x_L]_q, \cdots, 0)^T u_{pq}) + diag(\rho'(g_L)) \\
&W(\sum_{i,j} \frac{\partial h_{L-1}}{\partial w_{ij}} w_{ij} + \sum_{p,q} \frac{\partial h_{L-1}}{\partial u_{pq}} u_{pq}) \\
&= diag(\rho'(g_L))(W h_{L-1} + U x_L) + diag(\rho'(g_L)) W(\sum_{i,j} \frac{\partial h_{L-1}}{\partial w_{ij}} w_{ij} + \sum_{p,q} \frac{\partial h_{L-1}}{\partial u_{pq}} u_{pq}) \\
&= h_L + diag(\rho'(g_L)) W(\sum_{i,j} \frac{\partial h_{L-1}}{\partial w_{ij}} w_{ij} + \sum_{p,q} \frac{\partial h_{L-1}}{\partial u_{pq}} u_{pq}).
\end{aligned}$$

For ease of exposition, define $f_L := \sum_{i,j} \frac{\partial h_L}{\partial w_{ij}} w_{ij} + \sum_{p,q} \frac{\partial h_L}{\partial u_{pq}} u_{pq}$. Then the above equality can be rewritten as

$$f_L = h_L + diag(\rho'(g_L)) W f_{L-1}.$$

By induction, we have

$$\begin{aligned}
f_L = &h_L + diag(\rho'(g_L)) W h_{L-1} + diag(\rho'(g_L)) W diag(\rho'(g_{L-1})) W h_{L-2} + \\
&diag(\rho'(g_L)) W diag(\rho'(g_{L-1}))...W diag(\rho'(g_2)) W h_1
\end{aligned} \quad.$$

Multiplying both sides by $V$ and using some basic calculation, we can show that $V f_L = L y_L - \eta(\theta)(0, 1, \cdots, L-1)^T$. Therefore,

$$\sum_{a,b} \frac{\partial y_L}{\partial v_{ab}} v_{ab} + \sum_{i,j} \frac{\partial y_L}{\partial w_{ij}} w_{ij} + \sum_{p,q} \frac{\partial y_L}{\partial u_{pq}} u_{pq} = y_L + V f_L = (L+1) y_L - \eta(\theta)(0, 1, \cdots, L-1)^T \quad.$$

Substituting $y_L = \eta(\theta)(1, 1, \cdots, 1)^T$ into the above equality leads to the desired result

$$\sum_{a,b} \frac{\partial y_L}{\partial v_{ab}} v_{ab} + \sum_{i,j} \frac{\partial y_L}{\partial w_{ij}} w_{ij} + \sum_{p,q} \frac{\partial y_L}{\partial u_{pq}} u_{pq} = \eta(\theta) c \quad.$$

$\square$

## A.2 Proof of Lemma 2

*Proof.* Using the definition of Fisher-Rao norm,

$$||\theta||^2_{fr} = \mathbb{E}(<\theta, \nabla l(y_{L\theta}, y)>^2) = \mathbb{E}(<\theta, \nabla y_{L\theta}(x)\frac{\partial l(y_{L\theta}, y)}{\partial y_{L\theta}}>^2)$$
$$= \mathbb{E}((\theta^T \nabla y_{L\theta}(x)\frac{\partial l(y_{L\theta}, y)}{\partial y_{L\theta}})^2) = \mathbb{E}(<\nabla y_{L\theta}(x)^T\theta, \frac{\partial l(y_{L\theta}, y)}{\partial y_{L\theta}}>^2)$$

By Lemma 1, we have $\nabla y_{L\theta}(x)^T\theta = \eta(\theta)c$. Substituting it into the above equality gives us

$$||\theta||^2_{fr} = \mathbb{E}\left(\langle\eta(\theta)c, \frac{\partial l(y_{L\theta}(x), y)}{\partial y_{L\theta}}\rangle^2\right).$$

$\square$

# B Proofs in Section 3.2

## B.1 Proof of Lemma 3

The proof of Lemma 3 relies on the following result in Saniuk & Rhodes (1987).

**Proposition 1.** *Let X, Y$\in R^{n\times n}$ with Y symmetric and nonnegative definite. Then,*
$$trace(XY) \le ||X||_2 \cdot trace(Y).$$

Now we are ready to prove Lemma 3.

*Proof.* Denote $A := \mathbb{E}(((L+1)x_1^T, Lx_2^T, \cdots, 2x_L^T)^T((L+1)x_1^T, Lx_2^T, \cdots, 2x_L^T))$. By the definition of $||\theta||_{fs}$, for any $y \in \mathcal{Y}$, we have

$$||[\psi(\theta)]_y,{}^T||_A^2$$
$$= [\psi(\theta)]_y,\mathbb{E}(((L+1)x_1^T, Lx_2^T, \cdots, 2x_L^T)^T((L+1)x_1^T, Lx_2^T, \cdots, 2x_L^T))[\psi(\theta)]_y,{}^T$$
$$= \mathbb{E}([\psi(\theta)]_y,((L+1)x_1^T, Lx_2^T, \cdots, 2x_L^T)^T((L+1)x_1^T, Lx_2^T, \cdots, 2x_L^T)[\psi(\theta)]_y,{}^T)$$
$$= \mathbb{E}([(\tau(\theta)c)_y]^2) \le ||\theta||_{fs}^2$$

where $[\psi(\theta)]_y$, represents the y-*th* row of $\psi(\theta)$. On the other hand, from the definition of Rademacher complexities,

$$\hat{\mathfrak{R}}_n(\mathcal{F}_r) = \mathbb{E}_\sigma\left(\sup_{f\in\mathcal{F}_r}\frac{1}{n}\sum_{i=1}^n f(x_i)\sigma_i\right) = \mathbb{E}_\sigma\left(\sup_{\theta,y}\frac{1}{n}\sum_{i=1}^n[\psi(\theta)x_i]_y\sigma_i\right) = \mathbb{E}_\sigma\left(\sup_{\theta,y}\frac{1}{n}\sum_{i=1}^n[\psi(\theta)]_y,x_i\sigma_i\right)$$
$$= \mathbb{E}_\sigma\left(\sup_{\theta,y} < \frac{1}{n}\sum_{i=1}^n x_i\sigma_i, [\psi(\theta)]_y,{}^T >\right) \le \mathbb{E}_\sigma\left(\sup_{\theta,y}(||\frac{1}{n}\sum_{i=1}^n x_i\sigma_i||_{A^{-1}}||[\psi(\theta)]_y,{}^T||_A)\right)$$
$$\le r\mathbb{E}_\sigma(||\frac{1}{n}\sum_{i=1}^n x_i\sigma_i||_{A^{-1}}) = r\mathbb{E}_\sigma\sqrt{<(\frac{1}{n}\sum_{i=1}^n x_i\sigma_i)(\frac{1}{n}\sum_{i=1}^n x_i^T\sigma_i), A^{-1}>}$$
$$\le r\sqrt{\mathbb{E}_\sigma <(\frac{1}{n}\sum_{i=1}^n x_i\sigma_i)(\frac{1}{n}\sum_{i=1}^n x_i^T\sigma_i), A^{-1}>} = r\sqrt{\frac{1}{n^2}<\sum_{i=1}^n x_ix_i^T, A^{-1}>}$$
$$= \frac{r}{n}\sqrt{<XX^T, A^{-1}>} = \frac{r}{n}\sqrt{<XX^T, (C\mathbb{E}(xx^T)C)^{-1}>} = \frac{r}{n}\sqrt{trace(XX^TC^{-1}(\mathbb{E}(xx^T))^{-1}C^{-1})}$$

where the first inequality uses Cauchy-Schwarz inequality and $C := diag((L+1)I_d, LI_d, \cdots, 2I_d)$.

Since $C^{-1}$ and $\mathbb{E}(xx^T)$ are positive definite, we have

$$trace(XX^TC^{-1}(\mathbb{E}(xx^T))^{-1}C^{-1}) = trace(C^{-1}(\mathbb{E}(xx^T))^{-1}C^{-1}XX^T)$$
$$\le ||C^{-1}(\mathbb{E}(xx^T))^{-1}C^{-1}||_2 trace(XX^T) \le ||C^{-1}||_2||(\mathbb{E}(xx^T))^{-1}||_2||C^{-1}||_2 trace(XX^T)$$
$$= \frac{1}{4}||(\mathbb{E}(xx^T))^{-1}||_2||X||_F^2 \le \frac{1}{4}\frac{||X||_F^2}{\lambda_{min}(\mathbb{E}(xx^T))}$$

where the first inequality is by Proposition 1 and the last inequality uses $trace(XX^T) = ||X||_F^2$.

Therefore,

$$\hat{\mathfrak{R}}_n(\mathcal{F}_r) \le \frac{r||X||_F}{2n}\sqrt{\frac{1}{\lambda_{min}(\mathbb{E}(xx^T))}}.$$

$\square$

## B.2   PROOF OF LEMMA 4

*Proof.* Denote $H'_t := UX_t + WH'_{t-1}$ and $H'_1 := UX_1$. By the definition of $H_L$, we have

$$H_L - H'_L = \rho(UX_L + WH_{L-1}) - UX_L - WH'_{L-1}$$
$$= \rho(UX_L + WH_{L-1}) - UX_L - WH_{L-1} + W(H_{L-1} - H'_{L-1}) = H''_L + W(H_{L-1} - H'_{L-1}) \quad,$$

which by induction gives

$$H_L - H'_L = H''_L + WH''_{L-1} + \cdots + W^{L-1}(H_1 - H'_1) = \sum_{i=1}^{L} W^{L-i}H''_i \quad.$$

So the difference term can be rewritten as

$$VH_L - \psi(\theta)X = VH_L - VH'_L = \sum_{i=1}^{L} VW^{L-i}H''_i \quad,$$

where the second equality uses $\psi(\theta)X = VH'_L$. $\qquad\square$

## B.3   PROOF OF LEMMA 5

*Proof.* Using Riesz-Thorin Theorem, we have $||H''_t||_p \le ||H''_t||_1^{1/p}||H''_t||_\infty^{1-1/p}$. And since $H''_t = \rho(G_t) - G_t$, by the definition of the induced $L_1$ and $L_\infty$ matrix norm, we know $||H''_t||_1 \le ||G_t||_1$ and $||H''_t||_\infty \le ||G_t||_\infty$. Therefore,

$$||H''_t||_p \le ||H''_t||_1^{1/p}||H''_t||_\infty^{1-1/p} \le ||G_t||_1^{1/p}||G_t||_\infty^{1-1/p} \le m^{\frac{1}{p}(1-\frac{1}{p})}n^{\frac{1}{p}(1-\frac{1}{p})}||G_t||_p \quad,$$

where the last inequality uses some basic facts about matrix norm that $||G_t||_1 \le m^{1-1/p}||G_t||_p$ and $||G_t||_\infty \le n^{1/p}||G_t||_p$. $\qquad\square$

## B.4   PROOF OF LEMMA 6

*Proof.* For any $y \in \mathcal{Y}$, by Hölder's inequality, for any $p, q \ge 1$ with $\frac{1}{p} + \frac{1}{q} = 1$,

$$\frac{1}{n}[VW^{L-i}H''_i]_{y,}\sigma = \frac{1}{n}[V]_{y,}W^{L-i}H''_i\sigma \le \frac{1}{n}||H''^T_i W^{T^{L-i}}[V]_{y,}^T||_p||\sigma||_q$$
$$= ||[V]_{y,}^T||_p||W^T||_p^{L-i}||H''^T_i||_p n^{1/q-1} \le ||[V]_{y,}^T||_p||W^T||_p^{L-i}m^{\frac{1}{p}(1-\frac{1}{p})}n^{-\frac{1}{p^2}}||G_i^T||_p \quad.$$

In order to eliminate the dimension dependency on $m$ and simultaneously enjoy a faster convergence rate with respect to $n$, we choose $p = 1$. Then the above inequality reduces to $\frac{1}{n}[VW^{L-i}H''_i]_{y,}\sigma \le ||[V]_{y,}^T||_1||W^T||_1^{L-i}n^{-1}||G_i^T||_1 \le \beta_V\beta_W^{L-i}n^{-1}||G_i^T||_1 \le \beta_V\beta_W^{L-i}n^{-1}(\beta_U||X_i^T||_1 + \beta_W||H_{i-1}^T||_1)$. For $||H_{i-1}^T||_1$, we have

$$||H_{i-1}^T||_1 = ||\rho(X_{i-1}^T U^T + H_{i-1}^T W^T)||_1 \quad \begin{aligned} &\le ||X_{i-1}^T U^T + H_{i-2}^T W^T||_1 \\ &\le \beta_U||X_{i-1}^T||_1 + \beta_W||H_{i-2}^T||_1 \end{aligned} \quad,$$

which by induction gives

$$||H_{i-1}^T||_1 \le \beta_U \sum_{j=1}^{i-1} \beta_W^{i-1-j}||X_j^T||_1.$$

Therefore,

$$\mathbb{E}_\sigma\Big(\sup_{\theta\in\Omega, y\in\mathcal{Y}} \frac{1}{n}[VW^{L-i}H''_i]_{y,}\sigma\Big) \le \frac{1}{n}\beta_V\beta_U \sum_{j=1}^{i} \beta_W^{L-j}||X_j^T||_1.$$

$\qquad\square$

### B.5 PROOF OF THEOREM 1

*Proof.* Using the notations that we have introduced earlier, the empirical Rademacher complexity can be rewritten as

$$
\begin{aligned}
&\mathbb{E}_\sigma \big( \sup_{\theta \in \overline{\Omega}, y \in \mathcal{Y}} \frac{1}{n} \sum_{i=1}^n [y_{L\theta}(x_i)]_y \sigma_i \big) \\
&= \mathbb{E}_\sigma \big( \sup_{\theta \in \overline{\Omega}, y \in \mathcal{Y}} \frac{1}{n} [VH_L]_y, \sigma \big) \\
&= \mathbb{E}_\sigma \big( \sup_{\theta \in \overline{\Omega}, y \in \mathcal{Y}} \frac{1}{n} [\sum_{i=1}^L VW^{L-i} H_i'' + \psi(\theta)X]_y, \sigma \big) \\
&\le \sum_{i=1}^L \mathbb{E}_\sigma \big( \sup_{\theta \in \overline{\Omega}, y \in \mathcal{Y}} \frac{1}{n} [VW^{L-i} H_i'']_y, \sigma \big) + \mathbb{E}_\sigma \big( \sup_{\theta \in \overline{\Omega}, y \in \mathcal{Y}} \frac{1}{n} [\psi(\theta)X]_y, \sigma \big) \\
&\le \sum_{i=1}^L \mathbb{E}_\sigma \big( \sup_{\theta \in \Omega, y \in \mathcal{Y}} \frac{1}{n} [VW^{L-i} H_i'']_y, \sigma \big) + \mathbb{E}_\sigma \big( \sup_{||\theta||_{fs} \le r, y \in \mathcal{Y}} \frac{1}{n} [\psi(\theta)X]_y, \sigma \big)
\end{aligned}
$$,

where the second equality uses Lemma 4 and the last inequality is due to the fact that $\overline{\Omega} \subseteq \Omega$ and $\overline{\Omega} \subseteq \{\theta : ||\theta||_{fs} \le r\}$.

For the last term, by Lemma 3, we have

$$
\mathbb{E}_\sigma \big( \sup_{||\theta||_{fs} \le r, y \in \mathcal{Y}} \frac{1}{n} [\psi(\theta)X]_y, \sigma \big) \le \frac{r||X||_F}{2n} \sqrt{\frac{1}{\lambda_{min}(\mathbb{E}(xx^T))}} .
$$

The other terms can be handled by Lemma 6 in the following way.

$$
\begin{aligned}
\sum_{i=1}^L \mathbb{E}_\sigma \big( \sup_{\theta \in \Omega, y \in \mathcal{Y}} \frac{1}{n} [VW^{L-i} H_i'']_y, \sigma \big) &\le \sum_{i=1}^L \big( \tfrac{1}{n} \beta_V \beta_U \sum_{j=1}^i \beta_W^{L-j} ||X_j^T||_1 \big) \\
&\le \sum_{i=1}^L \big( \tfrac{1}{n} \beta_V \beta_U \sum_{j=1}^i \beta_W^{L-j} ||X^T||_1 \big) = \frac{1}{n} \frac{\beta_V \beta_U ||X^T||_1}{1 - \beta_W} \big( \frac{1 - \beta_W^L}{1 - \beta_W} - L\beta_W^L \big)
\end{aligned}
$$

for $\beta_W \ne 1$, and $\sum_{i=1}^L \mathbb{E}_\sigma \big( \sup_{\theta \in \Omega, y \in \mathcal{Y}} \frac{1}{n} [VW^{L-i} H_i'']_y \sigma \big) \le \frac{1}{n} \beta_V \beta_U B_2 \frac{L+L^2}{2}$ for $\beta_W = 1$, where the second inequality uses the definition of matrix norm $|| \cdot ||_1$.

Collecting all terms, we establish

$$
\mathbb{E}_\sigma \big( \sup_{\theta \in \overline{\Omega}, y \in \mathcal{Y}} \frac{1}{n} \sum_{i=1}^n [y_{L\theta}(x_i)]_y \sigma_i \big) \le \frac{r||X||_F}{2n} \sqrt{\frac{1}{\lambda_{min}(\mathbb{E}(xx^T))}} + \frac{1}{n} \beta_V \beta_U ||X^T||_1 \Lambda.
$$

$\square$

## C PROOFS IN SECTION 3.3

This section includes the full proofs of the generalization bound for training with random noise.

### C.1 LIPSCHITZ PROPERTIES OF RELU NONLINEARITIES AND MARGIN OPERATOR

We first establish the Lipschitz properties of the ReLU activation function and margin operator $\mathcal{M}(y_L(x), y) := \mathcal{M}_{y_L}(x, y)$.

**Lemma 9.** *Let $\rho : R^n \to R^n$ be the coordinate-wise ReLU function, then it is 1-Lipschitz according to $|| \cdot ||_p$ for any $p \ge 1$.*

*Proof.* For any $x, x' \in R^n$,

$$
||\rho(x) - \rho(x')||_p = \big( \sum |\rho(x)_i - \rho(x')_i|^p \big)^{1/p} \le \big( \sum |x_i - x_i'|^p \big)^{1/p} = ||x - x'||_p.
$$

$\square$

**Lemma 10.** *For every $j$ and every $p \geq 1$, $\mathcal{M}(\cdot, j)$ is 2-Lipschitz wrt $|| \cdot ||_p$.*

*Proof.* Let $y, y', j$ be given, and suppose that $\mathcal{M}(y, j) \leq \mathcal{M}(y', j)$ without loss of generality. Select coordinate $i \neq j$ so that $\mathcal{M}(y, j) = y_j - y_i$. Then

$$\mathcal{M}(y', j) - \mathcal{M}(y, j) = y'_j - \max_{l \neq j} y'_l - y_j + y_i \leq (y'_j - y_j) + (y_i - y'_i) \leq 2||y' - y||_\infty \leq 2||y - y'||_p.$$

$\square$

## C.2 PROOF OF LEMMA 8

*Proof.* We prove this Lemma by induction. Let $x = (x_1, x_2, \cdots, x_L)$ and $x' = (x'_1, x'_2, \cdots, x'_L)$. Denote $g'_t := Ux'_t + Wh'_{t-1}, h'_t := \rho(g'_t)$ and $y'_t := Vh'_t$. Then we have

$$||h_t - h'_t||_\infty = ||\rho(g_t) - \rho(g'_t)||_\infty \leq ||g_t - g'_t||_\infty = ||Ux_t + Wh_{t-1} - Ux'_t - Wh'_{t-1}||_\infty$$
$$\leq ||U^T||_1 ||x_t - x'_t||_\infty + ||W^T||_1 ||h_{t-1} - h'_{t-1}||_\infty,$$

where the first inequality uses Lemma 9 and the second inequality uses basic properties of $|| \cdot ||_\infty$. By induction, we get

$$||h_L - h'_L||_\infty \leq \sum_i ||U^T||_1 ||W^T||_1^{L-i} ||x_i - x'_i||_\infty.$$

Therefore,

$$||y_L - y'_L||_\infty = ||Vh_L - Vh'_L||_\infty \leq ||V||_\infty ||h_L - h'_L||_\infty \leq \sum_i ||V^T||_1 ||U^T||_1 ||W^T||_1^{L-i} ||x_i - x'_i||_\infty.$$

$\square$

## C.3 PROOF OF THEOREM 3

We begin by establishing two auxiliary lemmas that we will need for the subsequent theorem.

**Lemma 11.** *For every RNNs in (1) with weight matrices $\theta = (U, W, V)$, the following inequality holds for any $x = (x_1, x_2, \cdots, x_L)$ and $y$.*

$$|\mathbb{E}_\epsilon[\Phi_\alpha(\mathcal{M}_{y_L}(x, y)) - \Phi_\alpha(\mathcal{M}_{y_L}(x + \epsilon, y))]| \leq \frac{2}{\alpha} \sum_i ||V^T||_1 ||U^T||_1 ||W^T||_1^{L-i} (\mathbb{E}_\epsilon ||\epsilon_i||_\infty).$$

*Proof.* For any fixed $x$ and $y$,

$$|\mathbb{E}_\epsilon[\Phi_\alpha(\mathcal{M}_{y_L}(x, y)) - \Phi_\alpha(\mathcal{M}_{y_L}(x + \epsilon, y))]| \leq \mathbb{E}_\epsilon |\Phi_\alpha(\mathcal{M}_{y_L}(x, y)) - \Phi_\alpha(\mathcal{M}_{y_L}(x + \epsilon, y))|$$
$$\leq \frac{2}{\alpha} \mathbb{E}_\epsilon ||y_L(x) - y_L(x + \epsilon)||_\infty \leq \frac{2}{\alpha} \mathbb{E}_\epsilon (\sum_i ||V^T||_1 ||U^T||_1 ||W^T||_1^{L-i} ||\epsilon_i||_\infty)$$
$$= \frac{2}{\alpha} \sum_i ||V^T||_1 ||U^T||_1 ||W^T||_1^{L-i} (\mathbb{E}_\epsilon ||\epsilon_i||_\infty)$$

,

where the first inequality uses Jensen's inequality, the second inequality follows from the $\frac{1}{\alpha}$-Lipschitz property of $\Phi_\alpha(\cdot)$ and Lemma 10 and the last inequality is by Lemma 8. $\square$

**Lemma 12.** *Let $\{\epsilon_i\}_{i=1}^d$ be an i.i.d sequence of $\mathcal{N}(0, \sigma^2)$ variables, then $\mathbb{E}[\max_i |\epsilon_i|] \leq \sigma\sqrt{2\log(2d)}$.*

*Proof.* Define $Z = [\max_i |\epsilon_i|]$. For any $t > 0$, by Jensen' inequality, we have

$$e^{t\mathbb{E}(Z)} \leq \mathbb{E}(e^{tZ}) = \mathbb{E}(\max_i e^{t|\epsilon_i|}) \leq \sum_i \mathbb{E}(e^{t|\epsilon_i|}) = 2d\Phi(\sigma^2 t)e^{\sigma^2 t^2/2} \leq 2de^{\sigma^2 t^2/2},$$

where the second inequality uses the definition of normal distribution and $\Phi$ is the cumulative distribution function of the standard normal distribution. Taking $\log s$ on both sides and dividing by $t$, we get

$$\mathbb{E}(Z) \leq \frac{\log(2d)}{t} + \frac{\sigma^2 t}{2}.$$

Choosing $t = \dfrac{\sqrt{2log(2d)}}{\sigma}$ leads to the desired result,

$$\mathbb{E}(Z) \le \sigma\sqrt{2\log(2d)}.$$

$\square$

We now return to the proof of Theorem 3.

*Proof.* For any RNNs with weight matrices $\theta = (U, W, V)$ satisfying $||V^T||_1 \le \beta_V, ||W^T||_1 \le \beta_W, ||U^T||_1 \le \beta_U$, we have

$$
\begin{aligned}
&|\mathbb{E}_{x,y}[\Phi_\alpha(\mathcal{M}_{y_L}(x, y))] - \mathbb{E}_{x,\epsilon,y}[\Phi_\alpha(\mathcal{M}_{y_L}(x + \epsilon, y))| \\
&= |\mathbb{E}_{x,y}(\Phi_\alpha(\mathcal{M}_{y_L}(x, y)) - \mathbb{E}_\epsilon[\Phi_\alpha(\mathcal{M}_{y_L}(x + \epsilon, y))])| \\
&\le \mathbb{E}_{x,y}|\Phi_\alpha(\mathcal{M}_{y_L}(x, y)) - \mathbb{E}_\epsilon[\Phi_\alpha(\mathcal{M}_{y_L}(x + \epsilon, y))]| \le \frac{2}{\alpha}\sum_i \beta_V\beta_U\beta_W^{L-i}(\mathbb{E}_\epsilon||\epsilon_i||_\infty)
\end{aligned}
$$,

where the first equality is due to the fact that the input $x$ and noise $\epsilon$ are independent, the first inequality uses Jensen's inequality and the last inequality follows from Lemma 11. The inequality above can be rewritten as follows.

$$\mathbb{E}_{x,y}[\Phi_\alpha(\mathcal{M}_{y_L}(x, y))] \le \mathbb{E}_{x,\epsilon,y}[\Phi_\alpha(\mathcal{M}_{y_L}(x + \epsilon, y)) + \frac{2}{\alpha}\sum_i \beta_V\beta_U\beta_W^{L-i}(\mathbb{E}_\epsilon||\epsilon_i||_\infty) .$$

For the first term in the right hand side of the above inequality, by Theorem 2, with probability at least $1 - \delta$, the following holds:

$$
\begin{aligned}
&\mathbb{E}_{x,\epsilon,y}[\Phi_\alpha(\mathcal{M}_{y_L}(x + \epsilon, y)) \\
&\le \frac{4k}{\alpha}\Big(\frac{r||X + X_\epsilon||_F}{2n}\sqrt{\frac{1}{\lambda_{min}(\mathbb{E}((x + \epsilon)(x + \epsilon)^T))}} + \frac{1}{n}\beta_V\beta_U||X^T + X_\epsilon^T||_1\Lambda) + 3\sqrt{\frac{log\frac{2}{\delta}}{2n}} + \\
&\quad \frac{1}{n}\sum\Phi_\alpha(\mathcal{M}_{y_L}(x_i + \epsilon_i, y_i)) \\
&= \frac{4k}{\alpha}\Big(\frac{r||X + X_\epsilon||_F}{2n}\sqrt{\frac{1}{\lambda_{min}(\mathbb{E}(xx^T) + \mathbb{E}(\epsilon\epsilon^T))}} + \frac{1}{n}\beta_V\beta_U||X^T + X_\epsilon^T||_1\Lambda) + 3\sqrt{\frac{log\frac{2}{\delta}}{2n}} + \\
&\quad \frac{1}{n}\sum\Phi_\alpha(\mathcal{M}_{y_L}(x_i + \epsilon_i, y_i)) \\
&= \frac{4k}{\alpha}\Big(\frac{r||X + X_\epsilon||_F}{2n}\sqrt{\frac{1}{\lambda_{min}(\mathbb{E}(xx^T) + \sigma_\epsilon^2 I)}} + \frac{1}{n}\beta_V\beta_U||X^T + X_\epsilon^T||_1\Lambda) + 3\sqrt{\frac{log\frac{2}{\delta}}{2n}} + \\
&\quad \frac{1}{n}\sum\Phi_\alpha(\mathcal{M}_{y_L}(x_i + \epsilon_i, y_i)) \\
&= \frac{4k}{\alpha}\Big(\frac{r||X + X_\epsilon||_F}{2n}\sqrt{\frac{1}{\lambda_{min}(\mathbb{E}(xx^T)) + \sigma_\epsilon^2}} + \frac{1}{n}\beta_V\beta_U||X^T + X_\epsilon^T||_1\Lambda) + 3\sqrt{\frac{log\frac{2}{\delta}}{2n}} + \\
&\quad \frac{1}{n}\sum\Phi_\alpha(\mathcal{M}_{y_L}(x_i + \epsilon_i, y_i))
\end{aligned}
$$.

Combining the above two inequalities together leads to

$$
\begin{aligned}
&\mathbb{E}[\mathbb{1}_{\mathcal{M}_{y_L}(x,y)\le 0}] \le \mathbb{E}_{x,y}[\Phi_\alpha(\mathcal{M}_{y_L}(x, y))] \\
&\le \frac{1}{n}\sum\Phi_\alpha(\mathcal{M}_{y_L}(x_i + \epsilon_i, y_i)) + \frac{2}{\alpha}\sum_i \beta_V\beta_U\beta_W^{L-i}(\mathbb{E}_\epsilon||\epsilon_i||_\infty) + \\
&\quad \frac{4k}{\alpha}\Big(\frac{r||X + X_\epsilon||_F}{2n}\sqrt{\frac{1}{\lambda_{min}(\mathbb{E}(xx^T)) + \sigma_\epsilon^2}} + \frac{1}{n}\beta_V\beta_U||X^T + X_\epsilon^T||_1\Lambda) + 3\sqrt{\frac{log\frac{2}{\delta}}{2n}}
\end{aligned}
$$,

where the first inequality makes use of the fact that $\mathbb{1}_u \le \Phi_\alpha(u)$. Therefore, the desired result can be immediately obtained by substituting $\mathbb{E}_\epsilon||\epsilon_i||_\infty$ with $\sigma_\epsilon\sqrt{2\log(2d)}$ according to Lemma 12. $\square$

## D    SUPPLEMENTARY FIGURES

Figure 2 shows the behavior of generalization error for RNNs as the standard derivation of noise $\sigma_\epsilon$ varies for the sequence length $L = 200$ and $300$ trained on IMDB dataset. As in the body, increasing $\sigma_\epsilon$ will first improve the generalization error and then, after a certain point, harm the performance of RNNs.

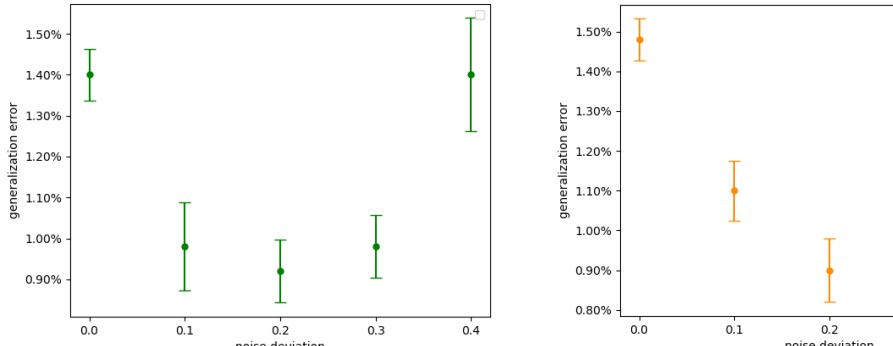

Figure 2: Generalization error for training with noise (mean $\pm$ standard error averaged on 5 runs). The left and right panel are for $L = 200$ (smallest eigenvalue: $1 \times 10^{-4}$) and $L = 300$ (smallest eigenvalue: $2.5 \times 10^{-5}$) respectively.

