# OpenReview forum: "Understanding Generalization in Recurrent Neural Networks"
_ICLR.cc/2020/Conference — Accept (Poster)_

### Official Review · AnonReviewer1 · 2019-10-22
**Official Blind Review #1**

**Rating:** 6

**Review:**

This paper proposes a new generalization bound for vanilla RNN with ReLU activation in terms of matrix-1 norm and Fisher-Rao norm. This bound has no explicit dependence on the size of networks.

I am actually not familiar with the generalization theorem on RNN. Nevertheless, according to the demonstration of the authors, I can understand the results of the theorems in this paper. I think the analysis on the property of the covariance matrix of the input data are valuable. However, I still have some concerns as follows.

1.	It is interesting if the bound can be independent on the size of networks. However, according to the bounds, I find that the bounds still depend on the size of the network but implicitly. Thus, I understand why the authors claim that the bound they provide has no “explicit” dependence on the size of networks. Then what is the value of this contribution?
2.	In Section 3.4.1, the authors state that “Since the smaller eigenvalues usually contribute to high frequency components of the input signal, our bound suggests that high frequency information is often more difficult to generalize, which is consistent with intuition.”. Can the authors provide more explanations on this since I do not understand why smaller eigenvalues usually contribute to high frequency components of the input signal and what the frequency components of the input signal is. What is the formulation of the high frequency components of the input signal?
3.	It seems that ReLU activation is not widely used in RNN. Instead, tangent and sigmoid function are more prevalent. The authors mention in Conclusion that the extending the results to other variants of RNNs like LSTM and MGU might be an interesting topic for future research. I think the first extending work is to study the generalization bound of vanilla RNN with tangent or sigmoid activation.



**Experience Assessment:**

I have read many papers in this area.

**Review Assessment: Checking Correctness Of Derivations And Theory:**

I assessed the sensibility of the derivations and theory.

**Review Assessment: Checking Correctness Of Experiments:**

I carefully checked the experiments.

**Review Assessment: Thoroughness In Paper Reading:**

I read the paper at least twice and used my best judgement in assessing the paper.

---

> ### Author Response · Authors · 2019-11-13
> **Response to Reviewer #1**
>
> We thank the reviewer for the comments and would like to address the reviewer’s concerns as follows.
>
> > I find that the bounds still depend on the size of the network but implicitly. Then what is the value of this contribution?
>
> Thanks. Our bound has no explicit dependence on the size of networks. But the norm itself used in our bound may still relate to the network size implicitly as the reviewer might concern. This is why we claim “explicit” in the paper. In fact, obtaining a generalization bound which is completely independent of the size of networks is an important future direction since such a result might answer a long standing question of why deep neural networks generalize so well in practice. Despite that it is still a long way to go, we already take a step further and give a bound without any explicit dependence on the network size by using a new proof technique in contrast to existing results which either have a factor of max{d,m,k} (Zhang et al., 2018) or the total number of parameters of the network (Chen et al., 2019) (see Comparison with existing results paragraph in Section 3.2). This new technique can potentially be applied to other neural networks architectures such as convolutional neural networks, which might be of independent interest. Besides, from our empirical results, the implicit dependency of norm on the network size might be much weaker than what we expect. For example, in our experiments, the matrix 1-norm of the weight matrix after training is around 2, which is much smaller than the size of matrix 128.
>
> > Why smaller eigenvalues usually contribute to high frequency components of the input signal and what the frequency components of the input signal is. What is the formulation of the high frequency components of the input signal?
>
> Thanks. Formally, the frequency component of an input signal can be defined via Fourier transform as $X(w) = \int_{-\infty}^{\infty} x(t) \cdot e^{-2 j \pi w t}dt$ where w stands for the frequency, x(t) is original input signal in time domain and X denotes the signal in frequency domain, and a signal x(t) has a high frequency component if X(w) has a relatively large value at a high frequency w. We refer the reviewer to [1] for more details.
>
> For eigenvalues, it corresponds to the variance explained by its eigenvectors. Not strictly speaking, the smaller eigenvalues contribute less to the energy in the data which usually correspond to the high frequency information, i.e., the details of an image. For example, in the application of Singular Value Decomposition to noise reduction in images, SVD can remove the noise and maintain most of the information in the original image by discarding those small singular values. Since most of the noise elements contribute to high frequency components of the image, we can think that the smaller eigenvalues correspond to the high frequency components and SVD acts like a low-pass filter.
>
> Their relationship might also be understood from a perspective of basis transformation. In short, both the eigenvectors and the Fourier transform can be seen as a change of basis. If we consider the eigenbasis of the space formed by all eigenvectors, we can show that the eigenbasis vectors corresponding to smaller eigenvalue are more sensitive to small changes in the data, i.e., a large condition number (Section 4.4, [3]). The same applies to the Hilbert space basis vectors $e^{j t w}$ with a high frequency w under the Fourier transform (Chapter 4, [2]). In this sense, smaller eigenvalues correspond to high frequency. But we have to mention that the true relationship between them might be much more complicated than what we claim in the paper. Therefore, for precision, we have replaced the word “usually” with “may” in the paper. We apologize for the misleading information and thank the reviewer again for pointing out this.
>
> [1]. R. Polikar, The Wavelet Tutorial. [Online]. Available: http://users.rowan.edu/~polikar/WTpart2.html
> [2]. Bertero, Mario, and Patrizia Boccacci. Introduction to inverse problems in imaging. CRC press, 1998.
> [3]. Workalemahu, Tsegaselassie. "Singular value decomposition in image noise filtering and reconstruction." (2008).
>
> >  I think the first extending work is to study the generalization bound of vanilla RNN with tangent or sigmoid activation.
>
> Thanks for the suggestion. For the future work, we agree that it is valuable to first extend our results to vanilla RNNs with tangent or sigmoid activation function.

---

### Official Review · AnonReviewer3 · 2019-10-23
**Official Blind Review #3**

**Rating:** 6

**Review:**

In this paper, the authors investigate the topic of theoretical generalizability in recurrent networks. Specifically, they extend a generalization bound using matrix 1-norm and the Fisher-Rao norm, proving the effectiveness of adding noise to input data for generalizability. These bounds have no dependence on the size of networks being investigated. The authors also propose a technique for representing RNNs as a decomposition into a sum of linear networks with a difference term, which allows for easier estimation of Rademacher complexity. The authors claim this is a representation that can be extended to other neural network architectures such as convolutional networks.

This work has the potential to be of interest for the learning theory community on theoretical properties of recurrent neural networks.

One question I have for the authors: in the experiments on the IMDB dataset, the authors claim that the generalization error is worst at \sigma_{\epsilon} = 0, but it appears that the error is actually larger for \sigma_{\epsilon} = 0.4?

**Experience Assessment:**

I do not know much about this area.

**Review Assessment: Checking Correctness Of Derivations And Theory:**

I did not assess the derivations or theory.

**Review Assessment: Checking Correctness Of Experiments:**

I assessed the sensibility of the experiments.

**Review Assessment: Thoroughness In Paper Reading:**

I made a quick assessment of this paper.

---

> ### Author Response · Authors · 2019-11-13
> **Response to Reviewer #3**
>
> We thank the reviewer for the comments. We have modified our manuscript to address your question.
>
> > In the experiments on the IMDB dataset, the authors claim that the generalization error is worst at $\sigma_{\epsilon} = 0$, but it appears that the error is actually larger for $\sigma_{\epsilon} = 0.4$?
>
> Thanks for pointing out this. We have changed the sentences to “We observe that as we start injecting noise, the generalization error becomes better. But when the deviation of noise keeps growing, the generalization error shows an increasing tendency”.

---

### Official Review · AnonReviewer4 · 2019-11-04
**Official Blind Review #4**

**Rating:** 6

**Review:**

This is an interesting and very well-written paper. I read the paper carefully but I don’t have sufficient expertise to determine whether all the proof steps are correct.

This paper builds on recent works giving generalization bounds on RNNs.

The primary contribution is that they are able to bound the generalization error for RNNs without a dependence on the network size parameters d and m.

Their proof is roughly as follows:
1. They decompose the RNN + ReLU activation into the sum of a linear network and difference terms. This step is key because it lets you treat each term independently when estimating the Rademacher complexity.
2. The linear network term can be bounded directly with their Fisher-Rao norm.
3. They make a second decomposition: the difference term can be written as a sum of simpler terms
4. For the simpler terms, their Rademacher complexity can be bounded independently using matrix-1 norm. Using the matrix-1 norm instead of the spectral norm means their bounds won’t depend on the network size parameters.
5. Then they combine these bounds to give the Rademacher complexity bounds for RNNs.
6. Lastly, they combine the Rademacher complexity bound with Kuznetsov et al 2015’s multiclass margin bound to give the generalization bound.

The downside to their generalization bound is that it requires the covariance matrix of the input data must be positive definitive, and it explodes when the smallest eigenvalue is close to zero.

Their second contribution attempts to address these downsides. They prove another generalization bound for RNNs when training with random noise, which has the effect of increasing the term containing the smallest eigenvalue of the input covariance matrix.

They remark on several empirical phenomena that are consistent with their results:
- Correlation of features in the input data makes it harder for RNNs to generalize
- Weight decay could help by decreasing the relevant gradient terms in their bounds
- Gradient clipping could help when the smallest eigenvalue of the input covariance is very small

Their third contribution is a single experiment. This contribution is fairly weak and the practical value of their theoretical work would be much more convincing if they were to put more effort into this section.
- They use IMDB data set (50k movie reviews + binary sentiment classification task)
- They add Gaussian noise to the input data with four different values.
- They plot the generalization error, which is the difference between the test error without noise and the training error with noise

Specific comments to improve their experiment section:
- I’m confused by the following sentence: “generalization errors … for different combinations of L and \sigma_epsilon are shown in Figure 1.” However, in Figure 1 I only see different values of sigma. I don’t see anything about using various values for L. This should be clarified.
- Figure 1 draws linear interpolation between data points - there’s no evidence for those interpolations. It should be reported as a scatter plot, preferably with error bars.
- If space limitations prevent reporting the experiment results more rigorously, I would prefer to see the experiment results reported in an appendix, with a brief comment on their significance in the main paper.

**Experience Assessment:**

I do not know much about this area.

**Review Assessment: Checking Correctness Of Derivations And Theory:**

I assessed the sensibility of the derivations and theory.

**Review Assessment: Checking Correctness Of Experiments:**

I carefully checked the experiments.

**Review Assessment: Thoroughness In Paper Reading:**

I read the paper thoroughly.

---

> ### Author Response · Authors · 2019-11-13
> **Response to Reviewer #4**
>
> Thank you for the detailed reviews. We have updated the experimental section according to your suggestions and answer specific questions below.
>
> > I am confused by the following sentence: “generalization errors … for different combinations of $L$ and $\sigma_\epsilon$ are shown in Figure 1.”
>
> We apologize for the confusion caused. We have changed the sentence to “… for $L=100$ and different values of $\sigma_\epsilon$ is shown in Figure 1 (results for other values of $L$ in Appendix D) ” and added more experiments for various values of $L$ in new Appendix D.
>
> >  The experimental results should be reported as a scatter plot, preferably with error bars.
>
> Thanks. We have replotted the empirical results with error bars in Figure 1.
>
> >  If space limitations prevent reporting the experimental results more rigorously, I would prefer to see the experiment results reported in an appendix.
>
> Thanks. We have added a new Appendix D which includes additional experimental results.

---

### Author Response · Authors · 2019-11-13
**Response to all reviewers**

We appreciate the insightful and constructive comments by all reviewers. We have revised the manuscript as suggested by the reviewers, and summarize the major changes as follows:

> Update all misleading sentences pointed out by the reviewers.

> Replot Figure 1 with error bars.

> Add a new Appendix D which includes additional experimental results.

Apart from the above changes, we have also fixed some typos. Please see answers to individual reviewer below, for particular comments. We thank the reviewers again for their effort in reviewing the manuscript.

---

### Decision · Program_Chairs · 2019-12-19

**Decision:**

Accept (Poster)

**Comment:**

This paper presents a generalization bound for RNNs based on matrix-1 norm and Fisher-Rao norm. As the initial bound relies on non-signularity of input covariance, which may not always hold in practice, the authors present additional analysis by noise injection to ensure covariance is positive definite. Through the resulted bound, the paper discusses how weight decay and gradient clipping in the training can help generalization. There were some concerns raised by reviewers, including  rigorous report of the experiment results,  claims on generalization in IMDB experiment,  claims of no explicit dependence on the size of networks, and the relationship of small eigenvalues in input covariance to high frequency features. The authors responded to these and also revised their draft to address most of these concerns (in particular, authors added a new section in the appendix that includes additional experimental results). Reviewers were mainly satisfied with the responses and the revision, and they all recommend accept.